# TopicRNN: A Recurrent Neural Network with Long-Range Semantic Dependency

**Adji B. Dieng** [*]
Columbia University
abd2141@columbia.edu

**Chong Wang**
Deep Learning Technology Center
Microsoft Research
chowang@microsoft.com

**Jianfeng Gao**
Deep Learning Technology Center
Microsoft Research
jfgao@microsoft.com

**John Paisley**
Columbia University
jpaisley@columbia.edu

## ABSTRACT

In this paper, we propose TopicRNN, a recurrent neural network (RNN)-based language model designed to directly capture the global semantic meaning relating words in a document via latent topics. Because of their sequential nature, RNNs are good at capturing the local structure of a word sequence – both semantic and syntactic – but might face difficulty remembering long-range dependencies. Intuitively, these long-range dependencies are of semantic nature. In contrast, latent topic models are able to capture the global semantic structure of a document but do not account for word ordering. The proposed TopicRNN model integrates the merits of RNNs and latent topic models: it captures local (syntactic) dependencies using an RNN and global (semantic) dependencies using latent topics. Unlike previous work on contextual RNN language modeling, our model is learned end-to-end. Empirical results on word prediction show that TopicRNN outperforms existing contextual RNN baselines. In addition, TopicRNN can be used as an unsupervised feature extractor for documents. We do this for sentiment analysis on the IMDB movie review dataset and report an error rate of $6.28\%$. This is comparable to the state-of-the-art $5.91\%$ resulting from a semi-supervised approach. Finally, TopicRNN also yields sensible topics, making it a useful alternative to document models such as latent Dirichlet allocation.

## 1   INTRODUCTION

When reading a document, short or long, humans have a mechanism that somehow allows them to remember the gist of what they have read so far. Consider the following example:

"*The U.S.presidential race isn't only drawing attention and controversy in the United States – it's being closely watched across the globe. But what does the rest of the world think about a campaign that has already thrown up one surprise after another? CNN asked 10 journalists for their take on the race so far, and what their country might be hoping for in America's next* —"

The missing word in the text above is easily predicted by any human to be either *President* or *Commander in Chief* or their synonyms. There have been various language models – from simple $n$-grams to the most recent RNN-based language models – that aim to solve this problem of predicting correctly the subsequent word in an observed sequence of words.

A good language model should capture at least two important properties of natural language. The first one is correct syntax. In order to do prediction that enjoys this property, we often only need to consider a few preceding words. Therefore, correct syntax is more of a local property. Word order matters in this case. The second property is the semantic coherence of the prediction. To achieve

---

[*]Work was done while at Microsoft Research.

this, we often need to consider many preceding words to understand the global semantic meaning of the sentence or document. The ordering of the words usually matters much less in this case.

Because they only consider a fixed-size context window of preceding words, traditional $n$-gram and neural probabilistic language models (Bengio et al., 2003) have difficulties in capturing global semantic information. To overcome this, RNN-based language models (Mikolov et al., 2010; 2011) use hidden states to "remember" the history of a word sequence. However, none of these approaches explicitly model the two main properties of language mentioned above, correct syntax and semantic coherence. Previous work by Chelba and Jelinek (2000) and Gao et al. (2004) exploit syntactic or semantic parsers to capture long-range dependencies in language.

In this paper, we propose TopicRNN, a RNN-based language model that is designed to directly capture long-range semantic dependencies via latent topics. These topics provide context to the RNN. Contextual RNNs have received a lot of attention (Mikolov and Zweig, 2012; Mikolov et al., 2014; Ji et al., 2015; Lin et al., 2015; Ji et al., 2016; Ghosh et al., 2016). However, the models closest to ours are the contextual RNN model proposed by Mikolov and Zweig (2012) and its most recent extension to the long-short term memory (LSTM) architecture (Ghosh et al., 2016). These models use pre-trained topic model features as an additional input to the hidden states and/or the output of the RNN. In contrast, TopicRNN does not require pre-trained topic model features and can be learned in an end-to-end fashion. We introduce an automatic way for handling stop words that topic models usually have difficulty dealing with. Under a comparable model size set up, TopicRNN achieves better perplexity scores than the contextual RNN model of Mikolov and Zweig (2012) on the Penn TreeBank dataset [1]. Moreover, TopicRNN can be used as an unsupervised feature extractor for downstream applications. For example, we derive document features of the IMDB movie review dataset using TopicRNN for sentiment classification. We reported an error rate of $6.28\%$. This is close to the state-of-the-art $5.91\%$ (Miyato et al., 2016) despite that we do not use the labels and adversarial training in the feature extraction stage.

The remainder of the paper is organized as follows: Section 2 provides background on RNN-based language models and probabilistic topic models. Section 3 describes the TopicRNN network architecture, its generative process and how to perform inference for it. Section 4 presents per-word perplexity results on the Penn TreeBank dataset and the classification error rate on the IMDB 100K dataset. Finally, we conclude and provide future research directions in Section 5.

## 2   BACKGROUND

We present the background necessary for building the TopicRNN model. We first review RNN-based language modeling, followed by a discussion on the construction of latent topic models.

### 2.1   RECURRENT NEURAL NETWORK-BASED LANGUAGE MODELS

Language modeling is fundamental to many applications. Examples include speech recognition and machine translation. A language model is a probability distribution over a sequence of words in a predefined vocabulary. More formally, let $V$ be a vocabulary set and $y_1, ..., y_T$ a sequence of $T$ words with each $y_t \in V$. A language model measures the likelihood of a sequence through a joint probability distribution,

$$p(y_1, ..., y_T) = p(y_1) \prod_{t=2}^{T} p(y_t|y_{1:t-1}).$$

Traditional $n$-gram and feed-forward neural network language models (Bengio et al., 2003) typically make Markov assumptions about the sequential dependencies between words, where the chain rule shown above limits conditioning to a fixed-size context window.

RNN-based language models (Mikolov et al., 2011) sidestep this Markov assumption by defining the conditional probability of each word $y_t$ given all the previous words $y_{1:t-1}$ through a hidden

---

[1] Ghosh et al. (2016) did not publish results on the PTB and we did not find the code online.

state $h_t$ (typically via a softmax function):

$$p(y_t|y_{1:t-1}) \triangleq p(y_t|h_t),$$
$$h_t = f(h_{t-1}, x_t).$$

The function $f(\cdot)$ can either be a standard RNN cell or a more complex cell such as GRU (Cho et al., 2014) or LSTM (Hochreiter and Schmidhuber, 1997). The input and target words are related via the relation $x_t \equiv y_{t-1}$. These RNN-based language models have been quite successful (Mikolov et al., 2011; Chelba et al., 2013; Jozefowicz et al., 2016).

While in principle RNN-based models can "remember" arbitrarily long histories if provided enough capacity, in practice such large-scale neural networks can easily encounter difficulties during optimization (Bengio et al., 1994; Pascanu et al., 2013; Sutskever, 2013) or overfitting issues (Srivastava et al., 2014). Finding better ways to model long-range dependencies in language modeling is therefore an open research challenge. As motivated in the introduction, much of the long-range dependency in language comes from semantic coherence, not from syntactic structure which is more of a local phenomenon. Therefore, models that can capture long-range semantic dependencies in language are complementary to RNNs. In the following section, we describe a family of such models called probabilistic topic models.

## 2.2 PROBABILISTIC TOPIC MODELS

Probabilistic topic models are a family of models that can be used to capture global semantic coherency (Blei and Lafferty, 2009). They provide a powerful tool for summarizing, organizing, and navigating document collections. One basic goal of such models is to find groups of words that tend to co-occur together in the same document. These groups of words are called topics and represent a probability distribution that puts most of its mass on this subset of the vocabulary. Documents are then represented as mixtures over these latent topics. Through posterior inference, the learned topics capture the semantic coherence of the words they cluster together (Mimno et al., 2011).

The simplest topic model is latent Dirichlet allocation (LDA) (Blei et al., 2003). It assumes $K$ underlying topics $\beta = \{\beta_1, \ldots, \beta_K\}$, each of which is a distribution over a fixed vocabulary. The generative process of LDA is as follows:
First generate the $K$ topics, $\beta_k \sim_{iid} \text{Dirichlet}(\tau)$. Then for each document containing words $y_{1:T}$, independently generate document-level variables and data:

1. Draw a document-specific topic proportion vector $\theta \sim \text{Dirichlet}(\alpha)$.

2. For the $t$th word in the document,

    (a) Draw topic assignment $z_t \sim \text{Discrete}(\theta)$.
    (b) Draw word $y_t \sim \text{Discrete}(\beta_{z_t})$.

Marginalizing each $z_t$, we obtain the probability of $y_{1:T}$ via a matrix factorization followed by an integration over the latent variable $\theta$,

$$p(y_{1:T}|\beta) = \int p(\theta) \prod_{t=1}^{T} \sum_{z_t} p(z_t|\theta) p(y_t|z_t, \beta) d\theta = \int p(\theta) \prod_{t=1}^{T} (\beta\theta)_{y_t} d\theta. \quad (1)$$

In LDA the prior distribution on the topic proportions is a Dirichlet distribution; it can be replaced by many other distributions. For example, the correlated topic model (Blei and Lafferty, 2006) uses a log-normal distribution. Most topic models are "bag of words" models in that word order is ignored. This makes it easier for topic models to capture global semantic information. However, this is also one of the reasons why topic models do not perform well on general-purpose language modeling applications such as word prediction. While bi-gram topic models have been proposed (Wallach, 2006), higher order models quickly become intractable.

Another issue encountered by topic models is that they do not model stop words well. This is because stop words usually do not carry semantic meaning; their appearance is mainly to make the sentence more readable according to the grammar of the language. They also appear frequently in

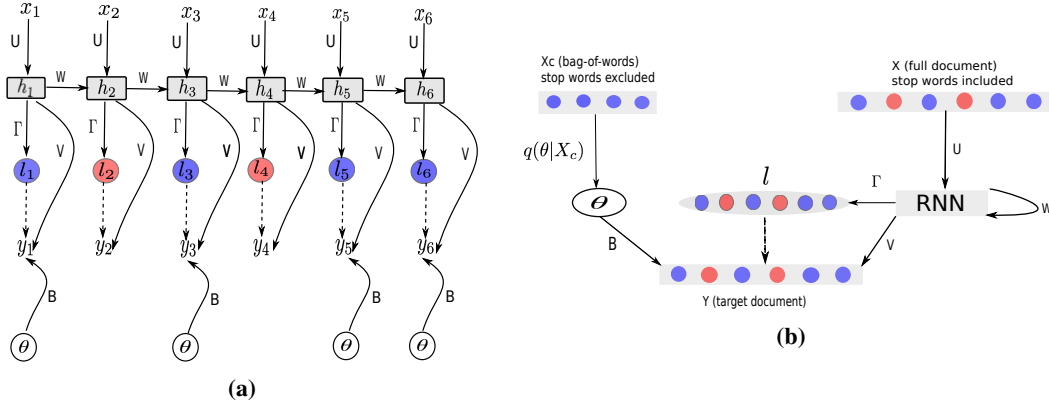

**Figure 1:** (a) The unrolled TopicRNN architecture: $x_1, ..., x_6$ are words in the document, $h_t$ is the state of the RNN at time step $t$, $x_i \equiv y_{i-1}$, $l_1, ..., l_6$ are stop word indicators, and $\theta$ is the latent representation of the input document and is unshaded by convention. (b) The TopicRNN model architecture in its compact form: $l$ is a binary vector that indicates whether each word in the input document is a stop word or not. Here red indicates stop words and blue indicates content words.

almost every document and can co-occur with almost any word[2]. In practice, these stop words are chosen using tf-idf (Blei and Lafferty, 2009).

## 3   THE TOPICRNN MODEL

We next describe the proposed TopicRNN model. In TopicRNN, latent topic models are used to capture global semantic dependencies so that the RNN can focus its modeling capacity on the local dynamics of the sequences. With this joint modeling, we hope to achieve better overall performance on downstream applications.

**The model.** TopicRNN is a generative model. For a document containing the words $y_{1:T}$,

1. Draw a topic vector[3] $\theta \sim N(0, I)$.
2. Given word $y_{1:t-1}$, for the $t$th word $y_t$ in the document,
   (a) Compute hidden state $h_t = f_W(x_t, h_{t-1})$, where we let $x_t \triangleq y_{t-1}$.
   (b) Draw stop word indicator $l_t \sim \text{Bernoulli}(\sigma(\Gamma^\top h_t))$, with $\sigma$ the sigmoid function.
   (c) Draw word $y_t \sim p(y_t | h_t, \theta, l_t, B)$, where

$$p(y_t = i | h_t, \theta, l_t, B) \propto \exp\left(v_i^\top h_t + (1 - l_t)b_i^\top \theta\right).$$

The stop word indicator $l_t$ controls how the topic vector $\theta$ affects the output. If $l_t = 1$ (indicating $y_t$ is a stop word), the topic vector $\theta$ has no contribution to the output. Otherwise, we add a bias to favor those words that are more likely to appear when mixing with $\theta$, as measured by the dot product between $\theta$ and the latent word vector $b_i$ for the $i$th vocabulary word. As we can see, the long-range semantic information captured by $\theta$ directly affects the output through an additive procedure. Unlike Mikolov and Zweig (2012), the contextual information is not passed to the hidden layer of the RNN. The main reason behind our choice of using the topic vector as bias instead of passing it into the hidden states of the RNN is because it enables us to have a clear separation of the contributions of global semantics and those of local dynamics. The global semantics come from the topics which are meaningful when stop words are excluded. However these stop words are needed for the local dynamics of the language model. We hence achieve this separation of global vs local via a binary decision model for the stop words. It is unclear how to achieve this if we pass the topics to the

---

[2]Wallach et al. (2009) described using asymmetric priors to alleviate this issue. Although it is not clear how to use this idea in TopicRNN, we plan to investigate such priors in future work.

[3]Instead of using the Dirichlet distribution, we choose the Gaussian distribution. This allows for more flexibility in the sequence prediction problem and also has advantages during inference.

hidden states of the RNN. This is because the hidden states of the RNN will account for all words (including stop words) whereas the topics exclude stop words.

We show the unrolled graphical representation of TopicRNN in Figure 1(a). We denote all model parameters as $\Theta = \{\Gamma, V, B, W, W_c\}$ (see Appendix A.1 for more details). Parameter $W_c$ is for the inference network, which we will introduce below. The observations are the word sequences $y_{1:T}$ and stop word indicators $l_{1:T}$.[4] The log marginal likelihood of the sequence $y_{1:T}$ is

$$\log p(y_{1:T}, l_{1:T}|h_t) = \log \int p(\theta) \prod_{t=1}^{T} p(y_t|h_t, l_t, \theta) p(l_t|h_t) \mathrm{d}\theta. \tag{2}$$

**Model inference.** Direct optimization of Equation 2 is intractable so we use variational inference for approximating this marginal (Jordan et al., 1999). Let $q(\theta)$ be the variational distribution on the marginalized variable $\theta$. We construct the variational objective function, also called the evidence lower bound (ELBO), as follows:

$$\mathcal{L}(y_{1:T}, l_{1:T}|q(\theta), \Theta) \triangleq \mathbb{E}_{q(\theta)} \left[ \sum_{t=1}^{T} \log p(y_t|h_t, l_t, \theta) + \log p(l_t|h_t) + \log p(\theta) - \log q(\theta) \right]$$
$$\leq \log p(y_{1:T}, l_{1:T}|h_t, \Theta).$$

Following the proposed variational autoencoder technique, we choose the form of $q(\theta)$ to be an inference network using a feed-forward neural network (Kingma and Welling, 2013; Miao et al., 2015). Let $X_c \in \mathcal{N}_+^{|V_c|}$ be the term-frequency representation of $y_{1:T}$ *excluding* stop words (with $V_c$ the vocabulary size without the stop words). The variational autoencoder inference network $q(\theta|X_c, W_c)$ with parameter $W_c$ is a feed-forward neural network with ReLU activation units that projects $X_c$ into a $K$-dimensional latent space. Specifically, we have

$$q(\theta|X_c, W_c) = N(\theta; \mu(X_c), \mathrm{diag}(\sigma^2(X_c))),$$
$$\mu(X_c) = W_1 g(X_c) + a_1,$$
$$\log \sigma(X_c) = W_2 g(X_c) + a_2,$$

where $g(\cdot)$ denotes the feed-forward neural network. The weight matrices $W_1$, $W_2$ and biases $a_1$, $a_2$ are shared across documents. Each document has its own $\mu(X_c)$ and $\sigma(X_c)$ resulting in a unique distribution $q(\theta|X_c)$ for each document. The output of the inference network is a distribution on $\theta$, which we regard as the summarization of the semantic information, similar to the topic proportions in latent topic models. We show the role of the inference network in Figure 1(b). During training, the parameters of the inference network and the model are jointly learned and updated via truncated backpropagation through time using the Adam algorithm (Kingma and Ba, 2014). We use stochastic samples from $q(\theta|X_c)$ and the reparameterization trick towards this end (Kingma and Welling, 2013; Rezende et al., 2014).

**Generating sequential text and computing perplexity.** Suppose we are given a word sequence $y_{1:t-1}$, from which we have an initial estimation of $q(\theta|X_c)$. To generate the next word $y_t$, we compute the probability distribution of $y_t$ given $y_{1:t-1}$ in an online fashion. We choose $\theta$ to be a point estimate $\hat{\theta}$, the mean of its current distribution $q(\theta|X_c)$. Marginalizing over the stop word indicator $l_t$ which is unknown prior to observing $y_t$, the approximate distribution of $y_t$ is

$$p(y_t|y_{1:t-1}) \approx \sum_{l_t} p(y_t|h_t, \hat{\theta}, l_t) p(l_t|h_t).$$

The predicted word $y_t$ is a sample from this predictive distribution. We update $q(\theta|X_c)$ by including $y_t$ to $X_c$ if $y_t$ is not a stop word. However, updating $q(\theta|X_c)$ after each word prediction is expensive, so we use a sliding window as was done in Mikolov and Zweig (2012). To compute the perplexity, we use the approximate predictive distribution above.

**Model Complexity.** TopicRNN has a complexity of $O(H \times H + H \times (C + K) + W_c)$, where $H$ is the size of the hidden layer of the RNN, $C$ is the vocabulary size, $K$ is the dimension of the topic vector, and $W_c$ is the number of parameters of the inference network. The contextual RNN of Mikolov and Zweig (2012) accounts for $O(H \times H + H \times (C + K))$, not including the pre-training process, which might require more parameters than the additional $W_c$ in our complexity.

---

[4]Stop words can be determined using one of the several lists available online. For example, `http://www.lextek.com/manuals/onix/stopwords2.html`

## 4   EXPERIMENTS

We assess the performance of our proposed TopicRNN model on word prediction and sentiment analysis[5]. For word prediction we use the Penn TreeBank dataset, a standard benchmark for assessing new language models (Marcus et al., 1993). For sentiment analysis we use the IMDB 100k dataset (Maas et al., 2011), also a common benchmark dataset for this application[6]. We use RNN, LSTM, and GRU cells in our experiments leading to TopicRNN, TopicLSTM, and TopicGRU.

**Table 1:** Five Topics from the TopicRNN Model with 100 Neurons and 50 Topics on the PTB Data. (The word *s&p* below shows as *sp* in the data.)

| Law | Company | Parties | Trading | Cars |
|---|---|---|---|---|
| law | spending | democratic | stock | gm |
| lawyers | sales | republicans | s&p | auto |
| judge | advertising | gop | price | ford |
| rights | employees | republican | investor | jaguar |
| attorney | state | senate | standard | car |
| court | taxes | oakland | chairman | cars |
| general | fiscal | highway | investors | headquarters |
| common | appropriation | democrats | retirement | british |
| mr | budget | bill | holders | executives |
| insurance | ad | district | merrill | model |

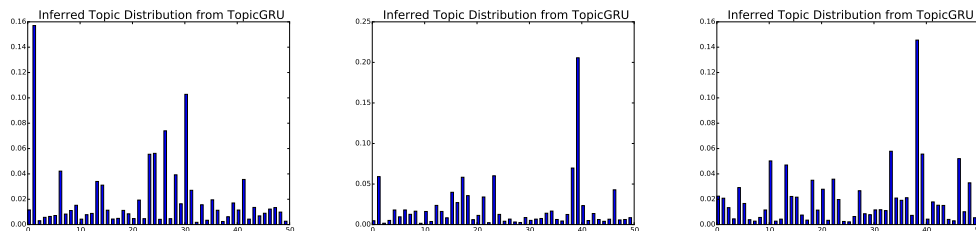

**Figure 2:** Inferred distributions using TopicGRU on three different documents. The content of these documents is added on the appendix. This shows that some of the topics are being picked up depending on the input document.

### 4.1   WORD PREDICTION

We first tested TopicRNN on the word prediction task using the Penn Treebank (PTB) portion of the Wall Street Journal. We use the standard split, where sections 0-20 (930K tokens) are used for training, sections 21-22 (74K tokens) for validation, and sections 23-24 (82K tokens) for testing (Mikolov et al., 2010). We use a vocabulary of size $10K$ that includes the special token *unk* for rare words and *eos* that indicates the end of a sentence. TopicRNN takes documents as inputs. We split the PTB data into blocks of 10 sentences to constitute documents as done by (Mikolov and Zweig, 2012). The inference network takes as input the bag-of-words representation of the input document. For that reason, the vocabulary size of the inference network is reduced to 9551 after excluding 449 pre-defined stop words.

In order to compare with previous work on contextual RNNs we trained TopicRNN using different network sizes. We performed word prediction using a recurrent neural network with 10 neurons,

---

[5]Our code will be made publicly available for reproducibility.

[6]These datasets are publicly available at http://www.fit.vutbr.cz/~imikolov/rnnlm/simple-examples.tgz and http://ai.stanford.edu/~amaas/data/sentiment/.

**Table 2:** TopicRNN and its counterparts exhibit lower perplexity scores across different network sizes than reported in Mikolov and Zweig (2012). Table 2a shows per-word perplexity scores for 10 neurons. Table 2b and Table 2c correspond to per-word perplexity scores for 100 and 300 neurons respectively. These results prove TopicRNN has more generalization capabilities: for example we only need a TopicGRU with 100 neurons to achieve a better perplexity than stacking 2 LSTMs with 200 neurons each: 112.4 vs 115.9)

**(a)**

| 10 Neurons | Valid | Test |
|---|---|---|
| RNN (no features) | 239.2 | 225.0 |
| RNN (LDA features) | 197.3 | 187.4 |
| TopicRNN | 184.5 | 172.2 |
| TopicLSTM | 188.0 | 175.0 |
| TopicGRU | 178.3 | **166.7** |

**(b)**

| 100 Neurons | Valid | Test |
|---|---|---|
| RNN (no features) | 150.1 | 142.1 |
| RNN (LDA features) | 132.3 | 126.4 |
| TopicRNN | 128.5 | 122.3 |
| TopicLSTM | 126.0 | 118.1 |
| TopicGRU | 118.3 | **112.4** |

**(c)**

| 300 Neurons | Valid | Test |
|---|---|---|
| RNN (no features) | — | 124.7 |
| RNN (LDA features) | — | 113.7 |
| TopicRNN | 118.3 | 112.2 |
| TopicLSTM | 104.1 | 99.5 |
| TopicGRU | 99.6 | **97.3** |

100 neurons and 300 neurons. For these experiments, we used a multilayer perceptron with 2 hidden layers and 200 hidden units per layer for the inference network. The number of topics was tuned depending on the size of the RNN. For 10 neurons we used 18 topics. For 100 and 300 neurons we found 50 topics to be optimal. We used the validation set to tune the hyperparameters of the model. We used a maximum of 15 epochs for the experiments and performed early stopping using the validation set. For comparison purposes we did not apply dropout and used 1 layer for the RNN and its counterparts in all the word prediction experiments as reported in Table 2. One epoch for 10 neurons takes 2.5 minutes. For 100 neurons, one epoch is completed in less than 4 minutes. Finally, for 300 neurons one epoch takes less than 6 minutes. These experiments were ran on Microsoft Azure NC12 that has 12 cores, 2 Tesla K80 GPUs, and 112 GB memory. First, we show five randomly drawn topics in Table 1. These results correspond to a network with 100 neurons. We also illustrate some inferred topic distributions for several documents from TopicGRU in Figure 2. Similar to standard topic models, these distributions are also relatively peaky.

Next, we compare the performance of TopicRNN to our baseline contextual RNN using perplexity. Perplexity can be thought of as a measure of surprise for a language model. It is defined as the exponential of the average negative log likelihood. Table 2 summarizes the results for different network sizes. We learn three things from these tables. First, the perplexity is reduced the larger the network size. Second, RNNs with context features perform better than RNNs without context features. Third, we see that TopicRNN gives lower perplexity than the previous baseline result reported by Mikolov and Zweig (2012). Note that to compute these perplexity scores for word prediction we use a sliding window to compute $\theta$ as we move along the sequences. The topic vector $\theta$ that is used from the current batch of words is estimated from the previous batch of words. This enables fair comparison to previously reported results (Mikolov and Zweig, 2012).[7]

Another aspect of the TopicRNN model we studied is its capacity to generate coherent text. To do this, we randomly drew a document from the test set and used this document as seed input to the inference network to compute $\theta$. Our expectation is that the topics contained in this seed document are reflected in the generated text. Table 3 shows generated text from models learned on the PTB and IMDB datasets. See Appendix A.3 for more examples.

---

[7]We adjusted the scores in Table 2 from what was previously reported after correcting a bug in the computation of the ELBO.

**Table 3:** Generated text using the TopicRNN model on the PTB (top) and IMDB (bottom).

*they believe that they had senior damages to guarantee and frustration of unk stations eos the rush to minimum effect in composite trading the compound base inflated rate before the common charter 's report eos wells fargo inc. unk of state control funds without openly scheduling the university 's exchange rate has been downgraded it 's unk said eos the united cancer & began critical increasing rate of N N at N N to N N are less for the country to trade rate for more than three months $ N workers were mixed eos*

*lee is head to be watched unk month she eos but the acting surprisingly nothing is very good eos i cant believe that he can unk to a role eos may appear of for the stupid killer really to help with unk unk unk if you wan na go to it fell to the plot clearly eos it gets clear of this movie 70 are so bad mexico direction regarding those films eos then go as unk 's walk and after unk to see him try to unk before that unk with this film*

**Table 4:** Classification error rate on IMDB 100k dataset. TopicRNN provides the state of the art error rate on this dataset.

| Model | Reported Error rate |
|---|---|
| BoW (bnc) (Maas et al., 2011) | 12.20% |
| BoW ($b\Delta$ tć) (Maas et al., 2011) | 11.77% |
| LDA (Maas et al., 2011) | 32.58% |
| Full + BoW (Maas et al., 2011) | 11.67% |
| Full + Unlabelled + BoW (Maas et al., 2011) | 11.11% |
| WRRBM (Dahl et al., 2012) | 12.58% |
| WRRBM + BoW (bnc) (Dahl et al., 2012) | 10.77% |
| MNB-uni (Wang & Manning, 2012) | 16.45% |
| MNB-bi (Wang & Manning, 2012) | 13.41% |
| SVM-uni (Wang & Manning, 2012) | 13.05% |
| SVM-bi (Wang & Manning, 2012) | 10.84% |
| NBSVM-uni (Wang & Manning, 2012) | 11.71% |
| seq2-bown-CNN (Johnson & Zhang, 2014) | 14.70% |
| NBSVM-bi (Wang & Manning, 2012) | 8.78% |
| Paragraph Vector (Le & Mikolov, 2014) | 7.42% |
| SA-LSTM with joint training (Dai & Le, 2015) | 14.70% |
| LSTM with tuning and dropout (Dai & Le, 2015) | 13.50% |
| LSTM initialized with word2vec embeddings (Dai & Le, 2015) | 10.00% |
| SA-LSTM with linear gain (Dai & Le, 2015) | 9.17% |
| LM-TM (Dai & Le, 2015) | 7.64% |
| SA-LSTM (Dai & Le, 2015) | 7.24% |
| **Virtual Adversarial (Miyato et al. 2016)** | **5.91%** |
| **TopicRNN** | **6.28%** |

## 4.2 SENTIMENT ANALYSIS

We performed sentiment analysis using TopicRNN as a feature extractor on the IMDB 100K dataset. This data consists of 100,000 movie reviews from the Internet Movie Database (IMDB) website. The data is split into 75% for training and 25% for testing. Among the 75K training reviews, 50K are unlabelled and 25K are labelled as carrying either a positive or a negative sentiment. All 25K test reviews are labelled. We trained TopicRNN on 65K random training reviews and used the remaining 10K reviews for validation. To learn a classifier, we passed the 25K labelled training reviews through the learned TopicRNN model. We then concatenated the output of the inference network and the last state of the RNN for each of these 25K reviews to compute the feature vectors. We then used these feature vectors to train a neural network with one hidden layer, 50 hidden units, and a sigmoid activation function to predict sentiment, exactly as done in Le and Mikolov (2014).

To train the TopicRNN model, we used a vocabulary of size 5,000 and mapped all other words to the *unk* token. We took out 439 stop words to create the input of the inference network. We used 500 units and 2 layers for the inference network, and used 2 layers and 300 units per-layer for the

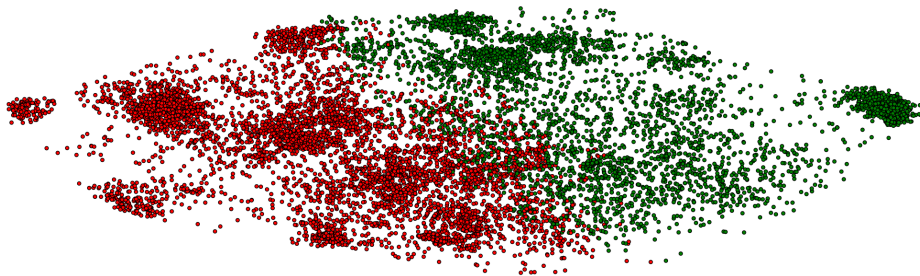

**Figure 3:** Clusters of a sample of 10000 movie reviews from the IMDB 100K dataset using TopicRNN as feature extractor. We used K-Means to cluster the feature vectors. We then used PCA to reduce the dimension to two for visualization purposes. red is a negative review and green is a positive review.

RNN. We chose a step size of 5 and defined 200 topics. We did not use any regularization such as dropout. We trained the model for 13 epochs and used the validation set to tune the hyperparameters of the model and track perplexity for early stopping. This experiment took close to 78 hours on a MacBook pro quad-core with 16GHz of RAM. See Appendix A.4 for the visualization of some of the topics learned from this data.

Table 4 summarizes sentiment classification results from TopicRNN and other methods. Our error rate is 6.28%.[8] This is close to the state-of-the-art 5.91% (Miyato et al., 2016) despite that we do not use the labels and adversarial training in the feature extraction stage. Our approach is most similar to Le and Mikolov (2014), where the features were extracted in a unsupervised way and then a one-layer neural net was trained for classification.

Figure 3 shows the ability of TopicRNN to cluster documents using the feature vectors as created during the sentiment analysis task. Reviews with positive sentiment are coloured in green while reviews carrying negative sentiment are shown in red. This shows that TopicRNN can be used as an unsupervised feature extractor for downstream applications. Table 3 shows generated text from models learned on the PTB and IMDB datasets. See Appendix A.3 for more examples. The overall generated text from IMDB encodes a negative sentiment.

## 5   DISCUSSION AND FUTURE WORK

In this paper we introduced TopicRNN, a RNN-based language model that combines RNNs and latent topics to capture local (syntactic) and global (semantic) dependencies between words. The global dependencies as captured by the latent topics serve as contextual bias to an RNN-based language model. This contextual information is learned jointly with the RNN parameters by maximizing the evidence lower bound of variational inference. TopicRNN yields competitive per-word perplexity on the Penn Treebank dataset compared to previous contextual RNN models. We have reported a competitive classification error rate for sentiment analysis on the IMDB 100K dataset. We have also illustrated the capacity of TopicRNN to generate sensible topics and text.
In future work, we will study the performance of TopicRNN when stop words are dynamically discovered during training. We will also extend TopicRNN to other applications where capturing context is important such as in dialog modeling. If successful, this will allow us to have a model that performs well across different natural language processing applications.

---

[8]The experiments were solely based on TopicRNN. Experiments using TopicGRU/TopicLSTM are being carried out and will be added as an extended version of this paper.

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

# A APPENDIX

## A.1 DIMENSION OF THE PARAMETERS OF THE MODEL:

We use the following notation: C is the vocabulary size (including stop words), H is the number of hidden units of the RNN, K is the number of topics, and E is the dimension of the inference network hidden layer. Table 5 gives the dimension of each of the parameters of the TopicRNN model (ignoring the biases).

**Table 5:** Dimensions of the parameters of the model.

|  | U | $\Gamma$ | W | V | B | $\theta$ | $W_1$ | $W_2$ |
|---|---|---|---|---|---|---|---|---|
| dimension | C x H | H | H x H | H x C | K x C | K | E | E |

## A.2    Documents used to infer the distributions on Figure 2

Figure on the left:    'the', 'market', 'has', 'grown', 'relatively', 'quiet', 'since', 'the', 'china', 'crisis', 'but', 'if', 'the', 'japanese', 'return', 'in', 'force', 'their', 'financial', 'might', 'could', 'compensate', 'to', 'some', 'extent', 'for', 'local', 'investors', "'", '<unk>', 'commitment', 'another', 'and', 'critical', 'factor', 'is', 'the', 'u.s.', 'hong', 'kong', "'s", 'biggest', 'export', 'market', 'even', 'before', 'the', 'china', 'crisis', 'weak', 'u.s.', 'demand', 'was', 'slowing', 'local', 'economic', 'growth', '<unk>', 'strong', 'consumer', 'spending', 'in', 'the', 'u.s.', 'two', 'years', 'ago', 'helped', '<unk>', 'the', 'local', 'economy', 'at', 'more', 'than', 'twice', 'its', 'current', 'rate', 'indeed', 'a', 'few', 'economists', 'maintain', 'that', 'global', 'forces', 'will', 'continue', 'to', 'govern', 'hong', 'kong', "'s", 'economic', '<unk>', 'once', 'external', 'conditions', 'such', 'as', 'u.s.', 'demand', 'swing', 'in', 'the', 'territory', "'s", 'favor', 'they', 'argue', 'local', 'businessmen', 'will', 'probably', 'overcome', 'their', 'N', 'worries', 'and', 'continue', 'doing', 'business', 'as', 'usual', 'but', 'economic', 'arguments', 'however', 'solid', 'wo', "n't", 'necessarily', '<unk>', 'hong', 'kong', "'s", 'N', 'million', 'people', 'many', 'are', 'refugees', 'having', 'fled', 'china', "'s", '<unk>', 'cycles', 'of', 'political', 'repression', 'and', 'poverty', 'since', 'the', 'communist', 'party', 'took', 'power', 'in', 'N', 'as', 'a', 'result', 'many', 'of', 'those', 'now', 'planning', 'to', 'leave', 'hong', 'kong', 'ca', "n't", 'easily', 'be', '<unk>', 'by', '<unk>', 'improvements', 'in', 'the', 'colony', "'s", 'political', 'and', 'economic', 'climate'

 Figure on the middle:    'it', 'said', 'the', 'man', 'whom', 'it', 'did', 'not', 'name', 'had', 'been', 'found', 'to', 'have', 'the', 'disease', 'after', 'hospital', 'tests', 'once', 'the', 'disease', 'was', 'confirmed', 'all', 'the', 'man', "'s", 'associates', 'and', 'family', 'were', 'tested', 'but', 'none', 'have', 'so', 'far', 'been', 'found', 'to', 'have', 'aids', 'the', 'newspaper', 'said', 'the', 'man', 'had', 'for', 'a', 'long', 'time', 'had', 'a', 'chaotic', 'sex', 'life', 'including', 'relations', 'with', 'foreign', 'men', 'the', 'newspaper', 'said', 'the', 'polish', 'government', 'increased', 'home', 'electricity', 'charges', 'by', 'N', 'N', 'and', 'doubled', 'gas', 'prices', 'the', 'official', 'news', 'agency', '<unk>', 'said', 'the', 'increases', 'were', 'intended', 'to', 'bring', '<unk>', 'low', 'energy', 'charges', 'into', 'line', 'with', 'production', 'costs', 'and', 'compensate', 'for', 'a', 'rise', 'in', 'coal', 'prices', 'in', '<unk>', 'news', 'south', 'korea', 'in', 'establishing', 'diplomatic', 'ties', 'with', 'poland', 'yesterday', 'announced', '$', 'N', 'million', 'in', 'loans', 'to', 'the', 'financially', 'strapped', 'warsaw', 'government', 'in', 'a', 'victory', 'for', 'environmentalists', 'hungary', "'s", 'parliament', 'terminated', 'a', 'multibillion-dollar', 'river', '<unk>', 'dam', 'being', 'built', 'by', '<unk>', 'firms', 'the', '<unk>', 'dam', 'was', 'designed', 'to', 'be', '<unk>', 'with', 'another', 'dam', 'now', 'nearly', 'complete', 'N', 'miles', '<unk>', 'in', 'czechoslovakia', 'in', 'ending', 'hungary', "'s", 'part', 'of', 'the', 'project', 'parliament', 'authorized', 'prime', 'minister', '<unk>', '<unk>', 'to', 'modify', 'a', 'N', 'agreement', 'with', 'czechoslovakia', 'which', 'still', 'wants', 'the', 'dam', 'to', 'be', 'built', 'mr.', '<unk>', 'said', 'in', 'parliament', 'that', 'czechoslovakia', 'and', 'hungary', 'would', 'suffer', 'environmental', 'damage', 'if', 'the', '<unk>', '<unk>', 'were', 'built', 'as', 'planned'

 Figure on the right:    'in', 'hartford', 'conn.', 'the', 'charter', 'oak', 'bridge', 'will', 'soon', 'be', 'replaced', 'the', '<unk>', '<unk>', 'from', 'its', '<unk>', '<unk>', 'to', 'a', 'park', '<unk>', 'are', 'possible', 'citizens', 'in', 'peninsula', 'ohio', 'upset', 'over', 'changes', 'to', 'a', 'bridge', 'negotiated', 'a', 'deal', 'the', 'bottom', 'half', 'of', 'the', '<unk>', 'will', 'be', 'type', 'f', 'while', 'the', 'top', 'half', 'will', 'have', 'the', 'old', 'bridge', "'s", '<unk>', 'pattern', 'similarly', 'highway', 'engineers', 'agreed', 'to', 'keep', 'the', 'old', '<unk>', 'on', 'the', 'key', 'bridge', 'in', 'washington', 'd.c.', 'as', 'long', 'as', 'they', 'could', 'install', 'a', 'crash', 'barrier', 'between', 'the', 'sidewalk', 'and', 'the', 'road', '<unk>', '<unk>', 'drink', 'carrier', 'competes', 'with', '<unk>', '<unk>', '<unk>', 'just', 'got', 'easier', 'or', 'so', 'claims', '<unk>', 'corp.', 'the', 'maker', 'of', 'the', '<unk>', 'the', 'chicago', 'company', "'s", 'beverage', 'carrier', 'meant', 'to', 'replace', '<unk>', '<unk>', 'at', '<unk>', 'stands', 'and', 'fast-food', 'outlets', 'resembles', 'the', 'plastic', '<unk>', 'used', 'on', '<unk>', 'of', 'beer', 'only', 'the', '<unk>', 'hang', 'from', 'a', '<unk>', 'of', '<unk>', 'the', 'new', 'carrier', 'can', '<unk>', 'as', 'many', 'as', 'four', '<unk>', 'at', 'once', 'inventor', '<unk>', 'marvin', 'says', 'his', 'design', 'virtually', '<unk>', '<unk>'

A.3    MORE GENERATED TEXT FROM THE MODEL:

We illustrate below some generated text resulting from training TopicRNN on the PTB dataset. Here we used 50 neurons and 100 topics:

Text1: *but the refcorp bond fund might have been unk and unk of the point rate eos house in national unk wall restraint in the property pension fund sold willing to zenith was guaranteed by $ N million at short-term rates maturities around unk products eos deposit posted yields slightly*

Text2: *it had happened by the treasury 's clinical fund month were under national disappear institutions but secretary nicholas instruments succeed eos and investors age far compound average new york stock exchange bonds typically sold $ N shares in the N but paying yields further an average rate of long-term funds*

We illustrate below some generated text resulting from training TopicRNN on the IMDB dataset. The settings are the same as for the sentiment analysis experiment:

*the film 's greatest unk unk and it will likely very nice movies to go to unk why various david proves eos the story were always well scary friend high can be a very strange unk unk is in love with it lacks even perfect for unk for some of the worst movies come on a unk gave a rock unk eos whatever let 's possible eos that kyle can 't different reasons about the unk and was not what you 're not a fan of unk unk us rock which unk still in unk 's music unk one as*

A.4    TOPICS FROM IMDB:

Below we show some topics resulting from the sentiment analysis on the IMDB dataset. The total number of topics is 200. Note here all the topics turn around movies which is expected since all reviews are about movies.

**Table 6:** Some Topics from the TopicRNN Model on the IMDB Data.

| pitt | tarantino | producing | ken | hudson | campbell | campbell |
|---|---|---|---|---|---|---|
| cameron | dramas | popcorn | opera | dragged | africa | spots |
| vicious | cards | practice | carrey | robinson | circumstances | dollar |
| francisco | unbearable | ninja | kong | flight | burton | cage |
| los | catches | cruise | hills | awake | kubrick | freeman |
| revolution | nonsensical | intimate | useless | rolled | friday | murphy |
| refuses | cringe | costs | lie | easier | expression | 2002 |
| cheese | lynch | alongside | repeated | kurosawa | struck | scorcese |

