# Peer review of "TopicRNN: A Recurrent Neural Network with Long-Range Semantic Dependency"

_ICLR 2017 — accepted_

[Official Review · AnonReviewer1 · rating 8 · confidence 4 · 17 Dec 2016]
**Nice work on feature extraction**

This work combines a LDA-type topic model with a RNN and models this by having an additive effect on the predictive distribution via the topic parameters. A variational auto-encoder is used to infer the topic distribution for a given piece of text and the RNN is trained as a RNNLM. The last hidden state of the RNNLM and the topic parameters are then concatenated to use as a feature representation.

The paper is well written and easy to understand. Using the topic as an additive effect on the vocabulary allows for easy inference but intuitively I would expect the topic to affect the dynamics too, e.g. the state of the RNN. The results on using this model as a feature extractor for IMDB are quite strong. Is the RNN fine-tuned on the labelled IMDB data? However, the results for PTB are weaker. From the original paper, an ensemble of 2 LSTMs is able to match the topicRNN score. This method of jointly modelling topics and a language model seems effective and relatively easy to implement.

Finally, the IMDB result is no longer state of the art since this result appeared in May (Miyato et al., Adversarial Training Methods for Semi-Supervised Text Classification).

Some questions:
How important is the stop word modelling? What do the results look like if l_t = 0.5 for all t?

It seems surprising that the RNN was more effective than the LSTM. Was gradient clipping tried in the topicLSTM case? Do GRUs also fail to work?

It is also unfortunate that the model requires a stop-word list. Is the link in footnote 4 the one that is used in the experiments?

Does factoring out the topics in this way allow the RNN to scale better with more neurons? How reasonable does the topic distribution look for individual documents? How peaked do they tend to be? Can you show some examples of the inferred distribution? The topics look odd for IMDB with the top word of two of the topics being the same: 'campbell'. It would be interesting to compare these topics with those inferred by LDA on the same datasets.

Minor comments:
Below figure 2: GHz -> GB
\Gamma is not defined.

[Official Review · AnonReviewer3 · rating 7 · confidence 4 · 19 Dec 2016]
originality 2 · clarity 3

This paper presents TopicRNN, a combination of LDA and RNN that augments traditional RNN with latent topics by having a switching variable that includes/excludes additive effects from latent topics when generating a word. 
Experiments on two tasks are performed: language modeling on PTB, and sentiment analysis on IMBD. 
The authors show that TopicRNN outperforms vanilla RNN on PTB and achieves SOTA result on IMDB.

Some questions and comments:
- In Table 2, how do you use LDA features for RNN (RNN LDA features)? 
- I would like to see results from LSTM included here, even though it is lower perplexity than TopicRNN. I think it's still useful to see how much adding latent topics close the gap between RNN and LSTM.
- The generated text in Table 3 are not meaningful to me. What is this supposed to highlight? Is this generated text for topic "trading"? What about the IMDB one?
- How scalable is the proposed method for large vocabulary size (>10K)?
- What is the accuracy on IMDB if the extracted features is used directly to perform classification? (instead of being passed to a neural network with one hidden state). I think this is a fairer comparison to BoW, LDA, and SVM methods presented as baselines.

[Official Review · AnonReviewer2 · rating 6 · confidence 3 · 20 Dec 2016]
**No Title**
clarity 3 · substance 3 · meaningful comparison 3

This paper introduces a model that blends ideas from generative topic models with those from recurrent neural network language models. The authors evaluate the proposed approach on a document level classification benchmark as well as a language modeling benchmark and it seems to work well. There is also some analysis as to topics learned by the model and its ability to generate text. Overall the paper is clearly written and with the code promised by the authors others should be able to re-implement the approach. I have 2 potentially major questions I would ask the authors to address:

1 - LDA topic models make an exchangability (bag of words) assumption. The discussion of the generative story for TopicRNN should explicitly discuss whether this assumption is also made. On the surface it appears it is since y_t is sampled using only the document topic vector and h_t but we know that in practice h_t comes from a recurrent model that observes y_t-1. Not clear how this clean exposition of the generative model relates to what is actually done. In the Generating sequential text section it’s clear the topic model can’t generate words without using y_1 - t-1 but this seems inconsistent with the generative model specification. This needs to be shown in the paper and made clear to have a complete paper.


2 -  The topic model only allows for linear interactions of the topic vector theta. It seems like this might be required to keep the generative model tractable but seems like a very poor assumption. We would expect the topic representation to have rich interactions with a language model to create nonlinear adjustments to word probabilities for a document. Please add discussion as to why this modeling choice exists and if possible how future work could modify that assumption (or explain why it’s not such a bad assumption as one might imagine)




Figure 2 colors very difficult to distinguish.

[Public Comment · (anonymous) · 30 Dec 2016]
**Is it unfair to use a global topic feature first and then do word prediction?**

I have a question regrading on the language modeling part. I believe it seems unfair to get a global word distribution(i.e. document topic) first and then use it to do word prediction. The RNN model would never do this and would perform not very good on the very beginning of this article. So does the ppl performance increase comes from this? 

What if the RNN model gets a global embedding first and then do the word prediction?

[Public Comment · Adji Bousso Dieng · 16 Jan 2017 (modified: 20 Jan 2017)]
**General Answer To Reviewers and ACs**

We thank the reviewers and the anonymous commenters for the helpful feedback and questions!
 
We first summarize the main idea of this paper below:
 
Neural network-based language models have achieved state of the art results on many NLP tasks. One difficult problem is to capture long-range dependencies as motivated in the introduction of this paper. We propose to solve this by integrating latent topics as context and jointly training these contextual features with the parameters of an RNN language model. We provide a natural way of doing this integration by modeling stop words that are excluded by topic models but needed for sequential language models. This is done via binary classification where the probability of being a stop word is dictated by the hidden layer of the RNN. This modeling approach is possible when the contextual features as provided by the topics are passed directly to the softmax output layer of the RNN as additional bias. We illustrate the performance of this approach on two tasks and two datasets: word prediction on PTB and sentiment analysis on IMDB.  We provide competitive perplexity scores on PTB showing more generalization capabilities (for example we only need a TopicGRU with 100 neurons to achieve a better perplexity than stacking 2 LSTMs with 200 neurons each ---112.4 vs 115.9). "This method of jointly modeling topics and a language model seems effective and relatively easy to implement." quoted from AnonReviewer1.
 
We have revised the paper and added the following changes:
1- we added a line on the middle of page 7 to clarify even more how we compute the topic vector \theta using a sliding window for word prediction.
2- we added the test perplexity scores for TopicRNN, TopicLSTM, and TopicGRU as required by AnonReviewer3.
3- we added the inferred distributions from some documents as required by AnonReviewer1.
4- we added an explanation of why we passed the topics directly to the output layer at the bottom of page 4. 

We answer each reviewer individually. See below.

[Final Decision · Program Chairs · 06 Feb 2017]
**ICLR committee final decision**

Though the have been attempts to incorporate both "topic-like" and "sequence-like" methods in the past (e.g, the work of Hanna Wallach, Amit Gruber and other), they were quite computationally expensive, especially when high-order ngrams are incorporated. This is a modern take on this challenge: using RNNs and the VAE / inference network framework. The results are quite convincing, and the paper is well written.
 
 Pros:
 -- clean and simple model
 -- sufficiently convincing experimentation
 
 Cons:
 -- other ways to model interaction between RNN and topic representation could be considered (see comments of R2 and R1)